# BANDWIDTH SELECTION FOR GAUSSIAN KERNEL RIDGE REGRESSION VIA JACOBIAN CONTROL

## ABSTRACT

Most machine learning methods require tuning of hyper-parameters. For kernel ridge regression with the Gaussian kernel, the hyper-parameter is the bandwidth. The bandwidth specifies the length scale of the kernel and has to be carefully selected in order to obtain a model with good generalization. The default methods for bandwidth selection are cross-validation and marginal likelihood maximization, which often yield good results, albeit at high computational costs. Furthermore, the estimates provided by these methods tend to have very high variance, especially when training data are scarce. Inspired by Jacobian regularization, we formulate an approximate expression for how the derivatives of the functions inferred by kernel ridge regression with the Gaussian kernel depend on the kernel bandwidth. We then use this expression to propose a closed-form, computationally feather-light, bandwidth selection heuristic, based on controlling the Jacobian. In addition, the Jacobian expression illuminates how the bandwidth selection is a trade-off between the smoothness of the inferred function and the conditioning of the training data kernel matrix. We show on real and synthetic data that compared to cross-validation and marginal likelihood maximization, our method is considerably faster and considerably more stable in terms of bandwidth selection.

## 1 INTRODUCTION

Kernel ridge regression, KRR, is a non-linear, closed-form solution regression technique used within a wide range of applications (Zahrt et al., 2019; Ali et al., 2020; Chen & Leclair, 2021; Fan et al., 2021; Le et al., 2021; Safari & Rahimzadeh Arashloo, 2021; Shahsavar et al., 2021; Singh Alvarado et al., 2021; Wu et al., 2021; Chen et al., 2022). It is related to Gaussian process regression (Krige, 1951; Matheron, 1963; Williams & Rasmussen, 2006), but with a frequentist, rather than a Bayesian, perspective. Apart from being useful on its own merits, in recent years, the similarities between KRR and neural networks have been highlighted, making the former an increasingly popular tool for gaining better theoretical understandings of the latter (Belkin et al., 2018; Jacot et al., 2018; Chen & Xu, 2020; Geifman et al., 2020; Ghorbani et al., 2020; 2021; Mei et al., 2021).

However, kernelization introduces hyper-parameters, which need to be carefully tuned in order to obtain good generalization. The bandwidth, $\sigma$, is a hyper-parameter used by many kernels, including the Gaussian, or radial basis function, kernel. The bandwidth specifies the length scale of the kernel. A kernel with a too small bandwidth will treat most new data as far from any training observation, while a kernel with a too large bandwidth will treat each new data point as basically equidistant to all training observations. None of these situations will result in good generalization.

The problem of bandwidth selection has been extensively studied for kernel density estimation, KDE, which is the basis for KDE-based kernel regression, such as the Nadaraya-Watson estimator (Nadaraya, 1964; Watson, 1964) and locally weighted regression (Cleveland & Devlin, 1988). Köhler et al. (2014) review existing methods for bandwidth selection for KDE-based kernel regression, methods that all make varyingly strong assumptions on the underlying data density and smoothness of the non-parametric regression model, and how the latter can be approximately estimated. On one end of the spectrum, cross-validation makes almost no assumptions on the underlying data structure, resulting in a very flexible, but computationally heavy, estimator, with high variance. On the other end, with strong assumptions on the underlying data structure, Silverman's rule of thumb, originally from 1986, (Silverman, 2018) is a computationally light estimator with low variance, but possibly far

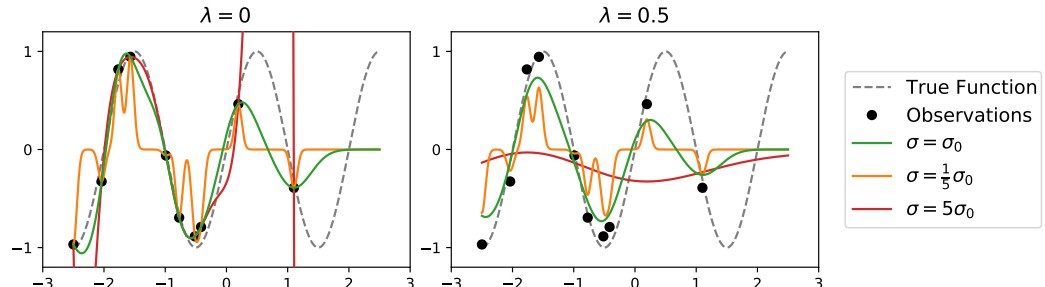

Figure 1: Kernel ridge regression with different bandwidths and different regularizations, where $\sigma_0$ is the bandwidth proposed by the Jacobian method, and $\lambda$ is the strength of the regularization. In the absence of regularization, regardless of the bandwidth, the inferred function perfectly interpolates the training data, i.e. it hits all training observations. When the bandwidth is too small, the kernel considers most new observations as far away from any training data and quickly resorts to its default value, 0. A too large bandwidth, on the other hand, results in extreme predictions between some of the observations. The addition of regularization affects larger bandwidths more than smaller ones. A too large bandwidth, in combination with regularization, produces a function that is too simple to capture the patterns in the training data.

from optimal. Other approaches on the spectrum include Park & Marron (1990), Sheather & Jones (1991), and Fan & Gijbels (1995).

Although similar in name and usage, KRR and KDE-based kernel regression are not the same. While KDE-based kernel regression estimates the probability density of the data, and uses this density to estimate $\mathbb{E}(y|\boldsymbol{x})$, KRR takes a functional perspective, directly estimating $\hat{y} = \hat{f}(\boldsymbol{x})$, similarly to how is done in neural networks.

For neural networks, Jacobian regularization, which penalizes the Frobenius norm of the Jacobian, $\left\|\frac{\partial \hat{f}(\boldsymbol{x})}{\partial \boldsymbol{x}}\right\|_F^2$, has recently been successfully applied to improve generalization (Jakubovitz & Giryes, 2018; Chan et al., 2019; Hoffman et al., 2019; Finlay et al., 2020; Bai et al., 2021). The Jacobian penalty is a non-linear generalization of the linear ridge penalty. To see this, consider the linear model $\hat{f}(\boldsymbol{x}) = \boldsymbol{x}^\top \boldsymbol{\beta}$, for which the Jacobian penalty becomes exactly the ridge penalty, $\|\boldsymbol{\beta}\|_2^2$. Thus, both Jacobian and ridge regularization improves generalization by constraining the derivatives of the inferred function.

This connection motivates our investigation into how Jacobian constraints can be applied for bandwidth selection in KRR: If we knew how the kernel bandwidth affects the Jacobian of the inferred function, then we could use Jacobian control as a criterion for selecting the bandwidth.

Our main contributions are:

- We derive an approximate expression for the Jacobian of the function inferred by KRR with the Gaussian kernel.
- We propose a closed-form, computationally feather-light, bandwidth selection method for KRR with the Gaussian kernel, based on controlling the approximate Jacobian.
- We show on synthetic and real data that Jacobian-based bandwidth selection outperforms cross-validation and marginal likelihood maximization in terms of computation speed and bandwidth selection stability.

## 2 BANDWIDTH SELECTION THROUGH JACOBIAN CONTROL

Consider the left panel of Figure 1. When large (absolute) derivatives of the inferred function are allowed, the function varies more rapidly between observations, while a function with constrained derivatives varies more smoothly, which intuitively improves generalization. However, the derivatives must not be too small as this leads to an overly smooth estimate, as seen in the right panel.

The functions in Figure 1 are all constructed using kernel ridge regression, KRR, with the Gaussian kernel. For training data $\boldsymbol{X} \in \mathbb{R}^{n \times p}$ and $\boldsymbol{y} \in \mathbb{R}^n$, the objective function of KRR is

$$\min_{f \in \mathcal{H}_k} \left\| \boldsymbol{y} - [f(\boldsymbol{x_1}) \quad \ldots \quad f(\boldsymbol{x_n})]^\top \right\|_2^2 + \lambda \|f\|_{\mathcal{H}_k}^2. \tag{1}$$

$\mathcal{H}_k$ denotes the reproducing kernel Hilbert space corresponding to the symmetric, positive semi-definite kernel function $k(\boldsymbol{x}, \boldsymbol{x'})$, and $\lambda \geq 0$ is the regularization strength. Solving Equation 1, a prediction $\hat{f}(\boldsymbol{x^*}) \in \mathbb{R}$, where $\boldsymbol{x^*} \in \mathbb{R}^p$, is given by

$$\hat{f}(\boldsymbol{x^*}) = \boldsymbol{k}\left(\boldsymbol{x^*}, \boldsymbol{X}\right)^\top \cdot \left(\boldsymbol{K}\left(\boldsymbol{X}, \boldsymbol{X}\right) + \lambda \boldsymbol{I}\right)^{-1} \cdot \boldsymbol{y}, \tag{2}$$

where $\boldsymbol{k}(\boldsymbol{x^*}, \boldsymbol{X}) \in \mathbb{R}^n$ and $\boldsymbol{K}(\boldsymbol{X}, \boldsymbol{X}) \in \mathbb{R}^{n \times n}$ are two kernel matrices, $\boldsymbol{k}(\boldsymbol{x^*}, \boldsymbol{X})_i = k(\boldsymbol{x^*}, \boldsymbol{x_i})$ and $\boldsymbol{K}(\boldsymbol{X}, \boldsymbol{X'})_{ij} = k(\boldsymbol{x_i}, \boldsymbol{x'_j})$.

The Gaussian kernel is given by

$$k_G(\boldsymbol{x}, \boldsymbol{x'}, \sigma) := \exp\left(-\frac{\|\boldsymbol{x} - \boldsymbol{x'}\|_2^2}{2\sigma^2}\right), \tag{3}$$

where the bandwidth, $\sigma$, specifies the length-scale of the kernel, i.e. what is to be considered as "close".

Returning to our aspiration from above, we would like to select $\sigma$ in Equation 3 to control $\left\| \frac{\partial \hat{f}(\boldsymbol{x^*})}{\partial \boldsymbol{x^*}} \right\|_F = \left\| \frac{\partial \hat{f}(\boldsymbol{x^*})}{\partial \boldsymbol{x^*}} \right\|_2$, with $\hat{f}(\boldsymbol{x^*})$ given by Equation 2.

In general, there is no simple expression for $\left\| \frac{\partial \hat{f}(\boldsymbol{x^*})}{\partial \boldsymbol{x^*}} \right\|_2$, but in Definition 2.1, we state an approximation that is based on derivations that we will present in Section 2.1.

**Definition 2.1** (Approximate Jacobian Norm)**.**

$$J_2^a(\sigma) = J_2^a(\sigma, l_{\max}, n, p, \lambda) := \frac{1}{\sigma} \cdot \frac{1}{n \cdot \exp\left(-\left(\frac{((n-1)^{1/p}-1)\pi\sigma}{2l_{\max}}\right)^2\right) + \lambda} \cdot C(n, \|\boldsymbol{y}\|_2), \tag{4}$$

where $l_{\max}$ denotes the maximum distance between two training observations, and $C(n, \|\boldsymbol{y}\|_2)$ is a constant with respect to $\sigma$.

**Remark 1**: Since we are only interested in how $J_2^a$ depends on $\sigma$, we will henceforth, with a slight abuse of notation, omit the constant $C(n, \|\boldsymbol{y}\|_2)$.

**Remark 2**: Technically, since we use a univariate response, $\frac{\partial \hat{f}(\boldsymbol{x^*})}{\partial \boldsymbol{x^*}}$ is a special case of the Jacobian, the gradient. We chose, however, to use the word Jacobian, since nothing in our derivations restricts us to the univariate case.

**Remark 3**: We will refer to the two $\sigma$ dependent factors in Equation 4 as

$$j_a(\sigma) := \frac{1}{\sigma} \quad \text{and} \quad j_b(\sigma) := \frac{1}{n \cdot \exp\left(-\left(\frac{((n-1)^{1/p}-1)\pi\sigma}{2l_{\max}}\right)^2\right) + \lambda}. \tag{5}$$

Proposition 2.2 below characterizes how the approximate Jacobian norm, $J_2^a$, depends on $\sigma$. Depending on $\lambda$, it can behave in three different ways: In the absence of regularization, $J_2^a$ becomes arbitrarily large for $\sigma$ small or large enough and enjoys a global minimum, $\sigma_0$, which is consistent with the left panel of Figure 1. However, as soon as regularization is added, $J_2^a$ goes to 0 as bandwidth goes to infinity, as indicated in the right panel. As long as the regularization parameter $\lambda \leq 2ne^{-3/2} \approx 0.45n$, there still exists a local minimum at $\sigma_0$. This is further illustrated in Figure 2, where we plot $J_2^a$ together with its components $j_a$ and $j_b$ for three different values of $\lambda$, reflecting the three types of behavior. Since $j_b$ is bounded by $1/\lambda$, for $\lambda$ large enough it is negligible compared to $j_a$.

**Proposition 2.2.**
*Let $J_2^a(\sigma)$ be defined according to Definition 2.1, and let, for $k \in \{-1, 0\}$,*

$$\sigma_k := \frac{\sqrt{2}}{\pi} \frac{l_{\max}}{(n-1)^{1/p} - 1} \sqrt{1 - 2W_k\left(-\frac{\lambda\sqrt{e}}{2n}\right)}, \tag{6}$$

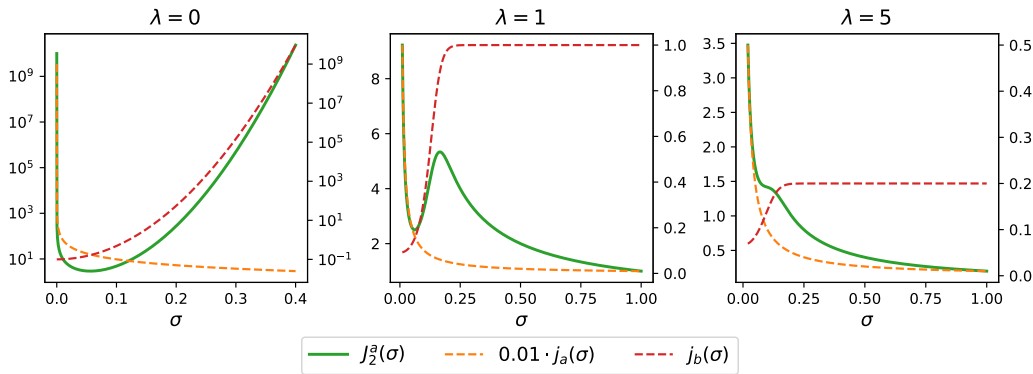

Figure 2: Approximate Jacobian norm, $J_2^a$, (left y-axis), and its two factors, $j_a$ and $j_b$, (right y-axis), as defined in Equations 4 and 5, as functions of the bandwidth for three different values of $\lambda$, and $n = 10$, $l_{\max} = 1$, $p = 1$. Note that the scales of the axes differ between the three panels and that $j_a(\sigma)$ is scaled down by a factor of 100. We clearly see the global and local minima stated in Proposition 2.2. For $\lambda = 0$, $J_2^a(0) = J_2^a(\infty) = +\infty$ with a global minimum at $\sigma_0$. For $\lambda > 0$, $J_2^a(0) = +\infty$ and $J_2^a(\infty) = 0$. For $\lambda \leq 2ne^{-3/2}$, $J_2^a$ has a local minimum at $\sigma_0$. $j_a(\sigma)$ decreases monotonically to 0, while $j_b(\sigma)$ increases monotonically to $1/\lambda$.

*where $W_k$ denotes the $k$-th branch of the Lambert W function. Then*

- *For $\lambda = 0$, $J_2^a(0) = J_2^a(\infty) = +\infty$, and $J_2^a(\sigma_0) = J_2^a\left(\frac{\sqrt{2}}{\pi} \frac{l_{\max}}{(n-1)^{1/p}-1}\right)$ is a global minimum.*

- *For $0 < \lambda \leq 2ne^{-3/2}$, $J_2^a(0) = +\infty$, and $J_2^a(\infty) = 0$, with a local minimum $J_2^a(\sigma_0)$ and a local maximum $J_2^a(\sigma_{-1})$.*

- *For $\lambda > 2ne^{-3/2}$, neither $\sigma_0$ nor $\sigma_{-1}$ is defined and $J_2^a(\sigma)$ decreases monotonically from $J_2^a(0) = +\infty$ to $J_2^a(\infty) = 0$.*

The proof is provided in the supplementary material.

**Remark**: In Figure 2, we provide examples of how $J_2^a(\sigma)$ changes with $\sigma$ for different values of $\lambda$, illustrating the minima and maxima stated in the proposition.

Based on Proposition 2.2 we can now propose a bandwidth selection scheme based on Jacobian control: For $\lambda \leq 2ne^{-3/2}$, we choose the (possibly local) minimum $\sigma_0$ as our Jacobian based bandwidth. For $\lambda > 2ne^{-3/2}$, $\sigma_0$ is not defined; in this case we choose our bandwidth as if $\lambda = 2ne^{-3/2}$. Note that $\sigma_0$ is quite stable to changes in $\lambda$: The square root expression in Equation 6 increases from 1 for $\lambda = 0$ to $\sqrt{3}$ for $\lambda = 2ne^{-3/2}$. This stability in terms of $\lambda$ can be seen in Figures 2 and 3.

## 2.1 THEORETICAL DETAILS

In this section, we present the calculations behind Definition 2.1. We also illuminate how bandwidth selection is a trade-off between a well-conditioned training data kernel matrix and a slow decay of the inferred function toward the default value.

We first use Proposition 2.3 to approximate the norm of the two kernel matrices in Equation 2 with a product of two matrix norms. We then use Propositions 2.4 and 2.5 to estimate these two norms for the case of the Gaussian kernel. Note that Proposition 2.3 holds for any kernel, not only the Gaussian.

**Proposition 2.3.**
*Let $d_i := x^* - x_i$ where $x_i$ is a row in $X$. Then, with $\hat{f}(x^*)$ according to Equation 2, for any*

*function $k(\boldsymbol{x}, \boldsymbol{x'})$,*

$$\left\|\frac{\partial \hat{f}(\boldsymbol{x^*})}{\partial \boldsymbol{x^*}}\right\|_2 = \left\|\frac{\partial \hat{f}(\boldsymbol{x^*})}{\partial \boldsymbol{d_i}}\right\|_2 \leq \max_{\boldsymbol{x_i} \in \boldsymbol{X}} \left\|\frac{\partial k(\boldsymbol{x^*}, \boldsymbol{x_i})}{\partial \boldsymbol{x^*}}\right\|_1 \cdot \left\|(\boldsymbol{K}(\boldsymbol{X}, \boldsymbol{X}) + \lambda \boldsymbol{I})^{-1}\right\|_2 \cdot \sqrt{n} \, \|\boldsymbol{y}\|_2$$
$$= \sqrt{n} \, \|\boldsymbol{y}\|_2 \cdot \max_{\boldsymbol{x_i} \in \boldsymbol{X}} \left\|\frac{\partial k(\boldsymbol{d_i} + \boldsymbol{x_i}, \boldsymbol{x_i})}{\partial \boldsymbol{d_i}}\right\|_1 \cdot \left\|(\boldsymbol{K}(\boldsymbol{X}, \boldsymbol{X}) + \lambda \boldsymbol{I})^{-1}\right\|_2, \tag{7}$$

*where the matrix norms are the induced operator norms.*

The proof is provided in the supplementary material.

**Proposition 2.4.**
*Let $\boldsymbol{d_i} := \boldsymbol{x^*} - \boldsymbol{x_i}$ where $\boldsymbol{x_i}$ is a row in $\boldsymbol{X}$, and denote $d_i := \|\boldsymbol{d_i}\|_2$. Then, for the Gaussian kernel, $k_G(\boldsymbol{d_i}, \sigma) = \exp\left(-\frac{\|\boldsymbol{d_i}\|_2^2}{2\sigma^2}\right)$,*

$$\max_{\boldsymbol{x_i} \in \boldsymbol{X}} \left\|\frac{\partial k_G(\boldsymbol{d_i}, \sigma)}{\partial \boldsymbol{d_i}}\right\|_1 = \max_{\boldsymbol{x_i} \in \boldsymbol{X}} \frac{d_i}{\sigma^2} \exp\left(-\frac{d_i^2}{2\sigma^2}\right) \leq \frac{1}{\sigma\sqrt{e}} =: \frac{1}{\sqrt{e}} \cdot j_a(\sigma). \tag{8}$$

The proof is provided in the supplementary material.

**Proposition 2.5.**
*For $\boldsymbol{K}(\boldsymbol{X}, \boldsymbol{X}, \sigma)_{ij} = k_G(\boldsymbol{x_i}, \boldsymbol{x_j}, \sigma)$, where $k_G(\boldsymbol{x}, \boldsymbol{x'}, \sigma)$ denotes the Gaussian kernel,*

$$\left\|(\boldsymbol{K}(\boldsymbol{X}, \boldsymbol{X}, \sigma) + \lambda \boldsymbol{I})^{-1}\right\|_2 \geq \frac{1}{n \cdot \exp\left(-\left(\frac{((n-1)^{1/P}-1)\pi\sigma}{2l_{\max}}\right)^2\right) + \lambda} =: j_b(\sigma). \tag{9}$$

The proof is provided in the supplementary material.

In the absence of regularization, $j_b(\sigma)$ is a bound of the spectral norm of the inverse training data kernel matrix. With increasing $\sigma$, the elements in $\boldsymbol{K}(\boldsymbol{X}, \boldsymbol{X}, \sigma)$ become increasingly similar, and $\boldsymbol{K}(\boldsymbol{X}, \boldsymbol{X}, \sigma)$ becomes closer to singular, which results in an ill-conditioned solution, where $\hat{f}(\boldsymbol{x^*})$ is very sensitive to perturbations in $\boldsymbol{X}$. Introducing regularization controls the conditioning, as seen in Figure 2; $j_b(\sigma)$ is upper bounded by $1/\lambda$. The poor generalization properties of regression with an ill-conditioned kernel matrix are well known, see e.g. Poggio et al. (2019), Amini (2021), or Hastie et al. (2022).

With the Jacobian approach, the contribution of an ill-conditioned matrix, $j_b(\sigma)$, is balanced by how quickly the inferred function decays in the absence of training data, $j_a(\sigma)$. For a too small bandwidth, the inferred function quickly decays to zero; $j_a(\sigma)$ is large and thus the derivatives of the inferred function. For a too large bandwidth, $\boldsymbol{K}(\boldsymbol{X}, \boldsymbol{X}, \sigma)$ is almost singular, which results in extreme predictions; $j_b(\sigma)$ is large, and thus the derivatives of the inferred function. By controlling the Jacobian, both poor generalization due to predicting mostly zero and poor generalization due to extreme predictions are avoided.

## 2.2 OUTLIER SENSITIVITY

Since $l_{\max}$ might be sensitive to outliers, Equation 6 suggests that so might the Jacobian method. One option to mitigate this problem is to use a trimmed version of $l_{\max}$, calculated after removing outliers. Our approach is however based on the observation that for data evenly spread within a hypercube in $\mathbb{R}^p$ with side $l_{\max}$, $\frac{l_{\max}}{(n-1)^{1/p}-1}$ is exactly the distance from an observation to its closest neighbor(s). We thus define the Jacobian median method analogously to the Jacobian method but with $\frac{l_{\max}}{(n-1)^{1/p}-1}$ replaced by $\mathrm{Med}_{i=1}^n \left(\min_{j \neq i} \|\boldsymbol{x_i} - \boldsymbol{x_j}\|_2\right)$, i.e. median of the closest-neighbor-distances.

## 3 EXPERIMENTS

Experiments were performed on three real and one synthetic data sets:

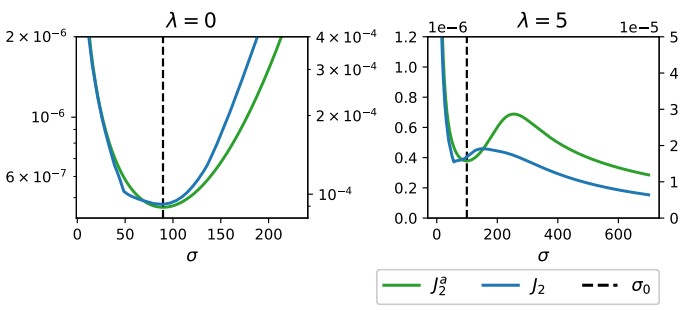
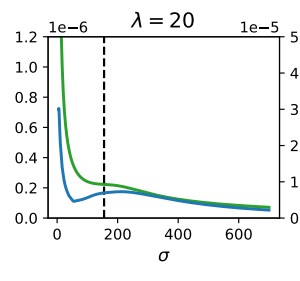

Figure 3: Comparison of the approximate (green, left y-axis) and true (blue, right y-axis) Jacobian norms for the 2D temperature data as a function of bandwidth and regularization. The approximate Jacobian norm captures the structure of the true Jacobian norm quite well. For $\lambda = 0$, the minima of the two functions agree very well, while for $\lambda > 0$, the selected bandwidth, $\sigma_0$, is close to the elbow of both the approximate and true norms. In the rightmost panel, $20 > 2 \cdot 40 \cdot e^{-3/2} = 17.85$, which means that the approximate Jacobian norm has no local minimum and $\sigma_0$ is selected as if $\lambda = 17.85$. See the supplementary material for corresponding comparisons on the other data sets.

- 2D Temperature Data: The temperatures at 40 different French weather stations at 3 a.m., Jan 1st, 2020.

- 1D Temperature Data: The temperature at 248 different times in January 2020 at the weather station at Toulouse-Blagnac.

- Cauchy Synthetic Data: For $n$ observations, for $i \in [1, n]$, $x_i \sim \text{Cauchy}(0, 3)$, $y_i = \sin(2\pi x_i) + \varepsilon_i$, where $\varepsilon_i \sim \mathcal{N}(0, 0.2^2)$.

- U.K. Rainfall Data: The rainfall at 164 monitoring stations in the U.K. every day in the year 2000.

The French temperature data was obtained from Météo France[1] and processed following the setup by Vanwynsberghe (2021). The U.K. rainfall data, used by Wood et al. (2017), was obtained from the author's homepage[2].

In all experiments, the Jacobian-based bandwidth was compared to that of generalized cross-validation (GCV) for kernel regression (Hastie et al., 2022) and marginal likelihood maximization (MML). For the likelihood maximization, Brent's method (Brent, 1971) was used, which is the default minimization method in Python's SciPy library (Virtanen et al., 2020). Unless otherwise stated, 100 logarithmically spaced values between 0.001 and $l_{\max}$ were used for the cross-validation. We also included Silverman's rule of thumb, $\sigma = \left(\frac{4}{n(p+2)}\right)^{1/(p+4)} \cdot \hat{\sigma}$, where $\hat{\sigma}$ is the estimated standard deviation of the observations. Note, however, that Silverman's method is developed with KDE, rather than KRR, in mind and thus does not take $\lambda$ into account. We chose however to include it as a reference since, just like the Jacobian method, it is a computationally light, closed-form solution. To avoid singular matrices, a small regularization of $\lambda = 10^{-3}$ was used in all experiments unless otherwise stated.

In Figure 3, we compare the approximate and true Jacobian norms for the 2D temperature data as a function of the bandwidth. The approximate norm captures the structure of the true norm quite well. In the absence of regularization, the minima of the two functions agree very well. When regularization is added the selected bandwidth, $\sigma_0$, is close to the elbow of both the approximate and true norms.

In Figure 6, we vary the sample size, $n$. For the synthetic data, 1000 test observations were generated, while the real data was randomly split into training and testing data. For each value of $n$, 15 % of the real data was saved for testing, resulting in splits of size $n/6$, $n/37$, and $n/1000$ for the 2D

---

[1] https://donneespubliques.meteofrance.fr/donnees_libres/Txt/Synop/postesSynop.csv

[2] https://www.maths.ed.ac.uk/~swood34/data/black_smoke.RData

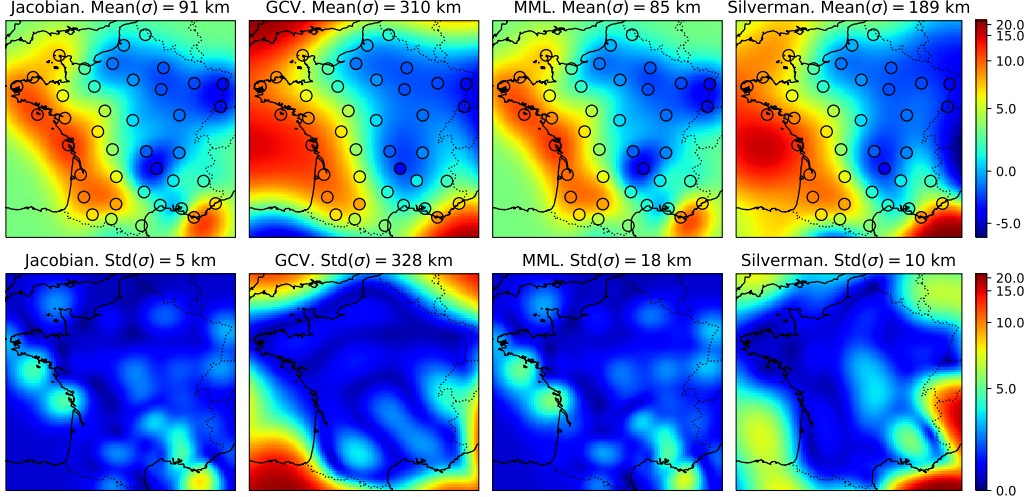

Figure 4: Means (top row) and standard deviations (bottom row) of KRR temperature predictions in $^\circ C$ from jackknife resampling on the 2D temperature data. Note that the scales are not linear. The Jacobian and MML methods are the most stable both in terms of predictions and bandwidth selection. They also have less extreme predictions.

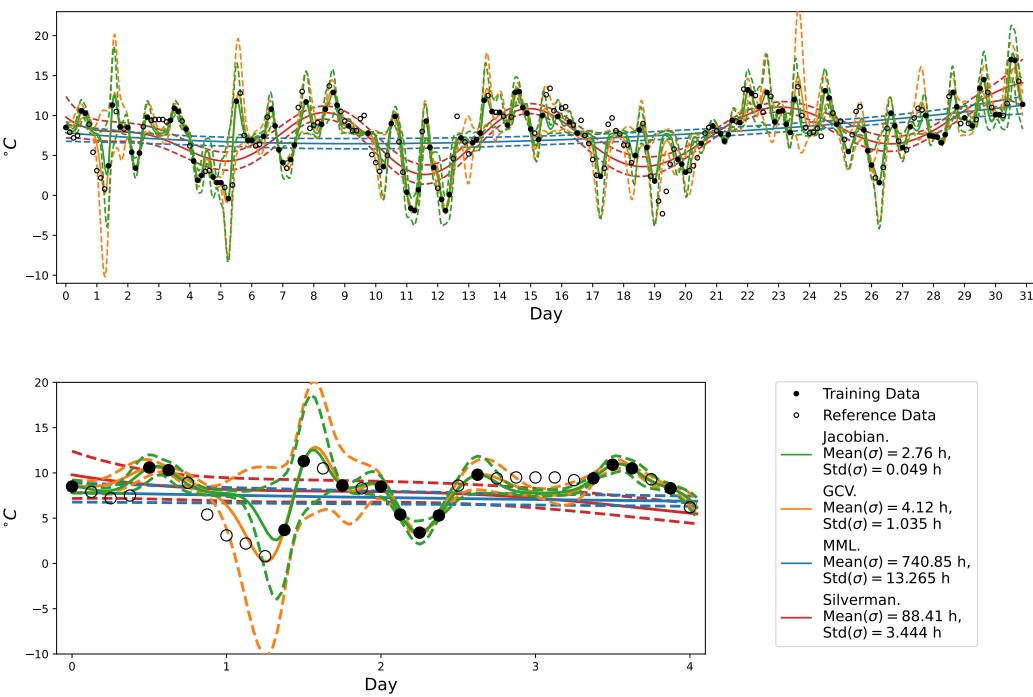

Figure 5: Means and standard deviations of KRR predictions from jackknife resampling on the 1D temperature data. The lower bottom plot shows a zoom-in on the first 4 days. The Jacobian method is more stable than cross-validation, both in terms of predictions and bandwidth selection. MML and Silverman's methods underfit the data, which can be attributed to their much larger bandwidth.

temperature, 1D temperature, and synthetic data, respectively. In all cases 1000 random splits were used to estimate the variance of $R^2$, i.e. the proportion of the variation in the data that is explained by $\hat{f}(\boldsymbol{x}^*)$, the selected bandwidth $\sigma$, and bandwidth selection computation time in milliseconds, $t$. The

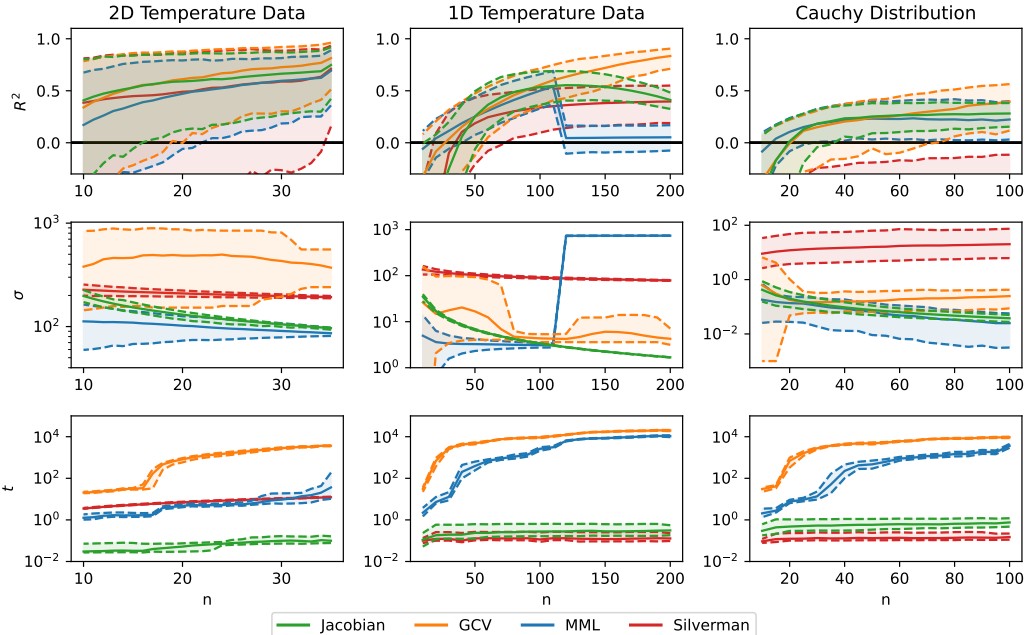

Figure 6: Mean together with first and ninth deciles for explained variance, $R^2$; selected bandwidth, $\sigma$; and computation time in milliseconds $t$, for different training sample sizes, using the four bandwidth selection methods. The Jacobian and Silverman's methods are several orders of magnitude quicker than the two other methods. They are also much more stable in terms of bandwidth selection. In terms of prediction, the Jacobian method generally performs better than, or on par with, the competing methods, except compared to cross-validation when $n$ is large. For the 1D temperature data, MML gets stuck in a local minimum. For the Cauchy data, the, slightly slower, median version of the Jacobian method was used. In the supplementary material, we include the corresponding figures when varying $\lambda$ for fixed $n$.

experiments were run on a cluster with Intel Xeon Gold 6130 CPUs. In the supplementary material, we include the corresponding experiments when varying $\lambda$ for fixed $n$.

It is again confirmed that the Jacobian method, in addition to being much faster than GCV and MML, is much more stable in terms of bandwidth selection. For the Cauchy distributed data, the median version of the Jacobian method was used; this method requires slightly more time than the standard Jacobian method. The reason for Silverman's method being slower than the Jacobian method for the 2D temperature data is due to its need to calculate the standard deviation of the data, and thus the distance to the mean from all observations. For the 2D temperature data, calculating the distances comprises a larger fraction of the calculations than for the other data. The other three methods do not use the standard deviation and are thus less affected.

We note that for the 1D temperature data, for large $n$, MML performs very badly, which can be attributed to a local minimum of the likelihood function. In the supplementary material, we investigate Jacobian seeded MML, where we use the Jacobian bandwidth as a seed, something that greatly improves the performance of MML.

To compare the methods on a massive data set, where computational time is crucial, we used the U.K. rainfall data. For each month the rainfall was predicted as a function of spatial coordinates and day, resulting in 12 experiments in total. The data, on average 4000 observations per month, was split randomly into 85 % training and 15 % testing data. For cross-validation, 10 logarithmically spaced values between $0.001$ and $l_{max}$ were used. The experiments were run on a Dell Latitude 7480 laptop, with an Intel Core i7, 2.80 GHz processor with four kernels. The results are presented in Table 1. For each method, we present the mean together with the first and ninth deciles for $R^2$, $\sigma$, and $t$. In addition, we include the results of Wilcoxon signed rank tests, testing whether the Jacobian method performs better (in terms of explained variance) and faster (in terms of computation time)

Table 1: Mean together with first and ninth deciles (within parentheses) of explained variance, $R^2$; bandwidth selection time in milliseconds, $t$; and selected bandwidth, $\sigma$, for the U.K. rainfall data. The Jacobian method performs significantly better and significantly faster than the competing methods. The p-value of 0.00024 corresponds to the Jacobian method performing better in all 12 experiments.

| Method | $R^2$ | $t$ [ms] | $\sigma$ |
|---|---|---|---|
| Jacobian | **0.981** (**0.967**, **0.994**) | **0.108** (**0.100**, **0.119**) | 0.214 (0.209, 0.219) |
| GCV | 0.969 (0.953, 0.988) $p_{\text{Wil}} = 0.00024$ | 25400 (20800, 28900) $p_{\text{Wil}} = 0.00024$ | 0.296 (0.131, 0.352) |
| MML | 0.554 (0.408, 0.678) $p_{\text{Wil}} = 0.00024$ | 29100 (24300, 33700) $p_{\text{Wil}} = 0.00024$ | 6.56 (6.47, 6.65) |
| Silverman | 0.971 (0.957, 0.990) $p_{\text{Wil}} = 0.00024$ | 0.168 (0.16, 0.18) $p_{\text{Wil}} = 0.00024$ | 0.306 (**0.304**, **0.309**) |

than the competing methods. The Jacobian method performs significantly better and is significantly faster than the competing methods. Compared to GCV and MML, the Jacobian method is more than a factor $10^5$ faster. The Jacobian method needs 0.1 milliseconds for bandwidth selection, to be compared with around half a minute for GCV and MML. Similarly to the 1D temperature data, unseeded MML uses a bandwidth that is too large.

## 4 CONCLUSIONS

We proposed a computationally low-cost method for choosing the bandwidth for kernel ridge regression with the Gaussian kernel. The method was motivated by the observed improved generalization properties of neural networks trained with Jacobian regularization. By selecting a bandwidth with Jacobian control, we implicitly constrain the derivatives of the inferred function. To achieve this, we derived a simple, closed-form approximate expression for the Jacobian of Gaussian KRR as a function of the bandwidth and were thus able to find an optimal bandwidth for Jacobian control. We demonstrated how selecting the optimal bandwidth is a trade-off between utilizing a well-conditioned training data kernel matrix and a slow decay of the inferred function toward the default value. Compared to cross-validation and maximal marginal likelihood estimation, apart from being considerably faster, and simpler to evaluate, our method is also more stable.

Even though we only considered Jacobian bandwidth selection for the Gaussian kernel, the principle holds for any kernel. That, however, requires new, kernel-specific, estimates of the kernel matrix norms. Similarly, in the Gaussian case, the estimate of the norm of the inverse kernel matrix could potentially be further improved. These research problems are left for future work.

Code is available in the supplementary material.

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
