# OpenReview forum: "Bandwidth Selection for Gaussian Kernel Ridge Regression via Jacobian Control"
_ICLR.cc/2024/Conference — ICLR 2024 Conference Withdrawn Submission_

### Official Review · Reviewer_Gr3G · 2023-10-27

**Soundness:** 2 fair
**Presentation:** 2 fair
**Contribution:** 2 fair
**Rating:** 5
**Confidence:** 3

**Summary:**

This paper proposes a Jacobian control method for selecting the bandwidth parameter of the Gaussian kernel in kernel ridge regression.
First, the authors derive an approximate expression for how the Jacobian norm of the inferred KRR function depends on the kernel bandwidth.
Then, based on controlling this approximate Jacobian, they propose a simple closed-form formula to select the bandwidth by minimizing the Jacobian norm. Finally, experiments on synthetic and real datasets show the Jacobian method is much faster than cross-validation and maximal likelihood for bandwidth selection; the Jacobian method is also more stable.

**Strengths:**

1. It is novel to introduce the Jacobian control method for choosing the hyperparameter in Gaussian kernel.
The closed form solution of the method results in significantly less computation time than the cross-validation and the maximal likelihood method, so the method is practical.
2. There are supporting experiments on both synthetic and real datasets. It is nice that the authors provide the codes of their experiments.

**Weaknesses:**

1. **The literature review in this paper should be improved.**
It seems that the authors only present some literature on the bandwidth selection for KDE, but do not review the literature for KRR (also, a distinction between KDE and KRR can be made clearer, since the 'kernel' here refer to different things).
Section 11.3 of the book [Support vector machines](https://link.springer.com/book/10.1007/978-0-387-77242-4) gives a kind of outdated overview, and more recent literature should be mentioned and discussed in this paper.

   In addition, works on the generalization performance of KRR should also be touched. For example, [Caponnetto 2007](https://link.springer.com/article/10.1007/s10208-006-0196-8), [Steinwart 2009](http://www.learningtheory.org/colt2009/papers/038.pdf), [Rakhlin 2018](http://arxiv.org/abs/1812.11167), [Li 2023](http://arxiv.org/abs/2303.15809).
   The very recent work [Haas 2023](http://arxiv.org/abs/2305.14077) also discussed the implications of derivates in KRR and a discussion would be interesting.

1. **The theoretical part is not convincing.** To derive the approximate Jacobian norm, the authors give an upper bound in (7) in terms of two factors. Then, they give an upper bound $j_a(\sigma)$ for the first term in (8) while a *lower bound* $j_b(\sigma)$ in (9) and claim that $J = j_a j_b$ is the approximation. Should there be an upper bound in (9) instead?

    Moreover, it would be more convincing that the authors provide some theoretical guarantee of this Jacobian method.


2. For the experiments, the Silverman’s rule of thumb method seems to be comparable to the Jacobian method (in Table 1).
More evidence could be provided to show the advantage of the Jacobian method.
Also, different choice of the regularization parameter $\lambda$ should be considered.

1. Although this paper proposes a method of choosing the bandwidth, the regularization parameter $\lambda$ is yet another hyperparameter in KRR. Using the same idea of Jacobian control method, it seems that there should be a uniform way to simultaneously choose $\lambda$ and $\sigma$.

**Minor**
1. I would recommend the authors to include pdf part of the supplementary material in the paper as appendix for readbility.

**Questions:**

* I would like the authors to respond to the points in the weaknesses part. I would like to increase my score if my concerns are addressed.

* In Table 1, how is the p-value computed? It seems that the p-values are all the same for different method?

---

### Official Review · Reviewer_q66a · 2023-10-30

**Soundness:** 2 fair
**Presentation:** 3 good
**Contribution:** 2 fair
**Rating:** 3
**Confidence:** 3

**Summary:**

In this works, the authors utilize the Jacobian regularization to consider the hyperparameter selection for Gaussian kernels in kernel ridge regresssion (KRR). The proposed method is efficient as a hyperparameter tuning method and is verfied with efficacy on the presented experiments, which are rather simple.

**Strengths:**

The topic studied in this work is interesting and is meaningful in kernel-based learning.

**Weaknesses:**

However, there are some concerns and confusions from the reviewer regarding the technical details and empirical practicality.

**Questions:**

### Major aspects:
1. In sec. 1, as far as the reviewer can see, the motivation of using this technique is simply bacause it gives some potentials in improving the generalization in neural networks, and then briefly mentions the special case of the linear ridge penalty. Then the authors claim that it motivates to apply this techinque to select the bandwidth in KRR. The reviewer finds it still unclear why it directly motivates to the selection of the bandwidth in KRR. Before this motivation, the reviewer would expect more explanations that can clearly connect the Jacobian regularization to Gaussian kernel and also the particular optimization problem of KRR, either in explanatory context or some empirical illustration.

2.  In definition 2.1, the $l_{max}$ is the maximal distance between two training observations. Are there some assumptions on the training data distribution? In proposition 2.5, it requires $l_{max}$ to compute the $j_b(\sigma)$. What would be the computational cost here to get the $l_{max}$ in practice, as it seems that paired-wise distances of two samples should be traversed? Though in sec 2.2., it mentions an alternative, but how good is this alternative compared to the original versions and how robust/sensitive are this alternative and the original version seems unexplored?

3.  As mentioned in the 2nd parragraph in sec 2.1, proposistion 2.4 and 2.5 are used to estimate the norms involved in (7). In (7), the upper bound is given, concerning the lhs of (8) and (9). However, (8) estimate an upper bound of lhs while (9) estimates a lower bound of lhs. Thus, in total in (7), it seems that the estimate of (7) through (8) and (9) is confusing.

4.  To the reviewer's understandings, the estimate of the bandwidth seems relevant to the regularization term $\lambda$? So, does it mean that the quality of the estimated bandwidth depends on the $\lambda$? In this why we should determine $\lambda$ first and estimate the bandwidth, and then repeat until a good pair of $\lambda$ and the bandwidth is finally determined? It would be more clear to have a clear algorithm step in the paper, as there is still some space.

5.  As the proposed Jacobina-regularization based method is motivated heuristically and it also involves some approximation to the gradient norm of KRR, comprehensive experiments should be presentd to verify the effectiveness. By far the presented experiments are rather too simple in sample size and dimensionality; besides the tested datasets are too few. In complicated task, even with a very well-selected bandwidth, the model may still lacks flexibility by only replying on the scalar of the bandwidth, so the reviewer wonders how the propoposed method can provides a better selection under such scenarios.


### Minors:
1. Tables of datasets descriptions can be given.
2. The information conveyed by fig.2 and fig.3 seems very alike.
3. In the appendix, it is better to align the numbering of propositions in the proofs with the main context, if possible.
4. Some other reviewed methods in sec.1 can be compared, if fair setups are set.

---

### Official Review · Reviewer_oRtQ · 2023-11-03

**Soundness:** 2 fair
**Presentation:** 3 good
**Contribution:** 2 fair
**Rating:** 3
**Confidence:** 4

**Summary:**

The manuscript proposes a methodology for the efficient selection of the bandwith/lengthscale parameter in kernel ridge regression with squared exponential kernel. The idea is to use an approximation to the norm of the Jacobian of the predictive mean and to select the lengthscale with minimal approximate Jacobian norm. Some experiments illustrate the applicability in practice.

**Strengths:**

* As the code to run the experiments is included in the submission and the datasets are public, it should be rather simple to reproduce the results.
* The proposed direct optimisation approach seems to be novel and is computationally lightweight.

**Weaknesses:**

* The empirical analysis is rather basic. The manuscript presents some results on small-scale datasets where the power of the method is potentially not required as a parameter grid search is simple to do.
 * The method -- as proposed -- is specific to the lengthscale estimation for kernel ridge regression with Gaussian covariance function.
 * The computational benefit is proven for small scale datasets only.

**Questions:**

* How can you conclude from Figure 3 rightmost panel that the approximation is "quite good" if the local minimum vanishes once lambda exceeds some threshold?
 * Can you improve Figure 2 in a way that the three cases of Proposition 2.2 can be clearly appreciated?
 * Can you make explicit whether the median scheme you propose in Section 2.2 is used in the experiments? Also, please use the same estimate also in Figure 3 for reasons of consistency.
 * How do you select the regularisation strength lambda? Why is it not optimised? How would this optimisation be interleaved with the estimation of sigma?
 * Is there a way to extend the methodology to optimise lambda, as well?
 * Is there a way to perform ARD with the proposed methodology?
 * Do you have an explanation of the weird behaviour of MML i.e. the much larger value of the estimated lengthscale on the 1D Temperature data shown in Figures 5+6 for n>110?
 * Can you please add an experiment comparing Jacobian median versus Jacobian max?
 * Can you please add an experiment using the exact Jacobian norm rather than the approximation to assess the impact of the approximation beyond visual comparison?

---

### Official Review · Reviewer_MVeD · 2023-11-08

**Soundness:** 1 poor
**Presentation:** 3 good
**Contribution:** 1 poor
**Rating:** 3
**Confidence:** 4

**Summary:**

The paper proposes a new method for choosing the length scale parameter in kernel ridge regression. The authors derive an approximation of the norm of the derivative of the KRR prediction (or the norm of the Jacobian if the input dimension is greater than 1), and propose to choose the length scale as a global/local minimum of this approximation. One notable point is that the proposed estimation procedure does not take into account the observed values $y_i$.

The experimental section of the paper illustrates the method on three real and one synthetic datasets, and the performance is compared to cross validation and maximum likelihood estimation. The authors show a good behaviour of the proposed approach on three metrics: the computation time associated with estimating the length scale, the variability of the estimate, and the predictive performance of the resulting model (measured by $R^2$).

**Strengths:**

One strength of the paper lies in the problem it aims at solving: Training KRR/Gaussian process models is typically a computationally demanding task and finding cheap yet good methods for parameter estimation would be of great benefit for the ICLR community.

Although some minor improvement are possible regarding clarity, the overall paper is easy to follow and clearly written.

I also appreciated that the authors included the code as supplementary material to facilitate the reproducibility of the results they present.

**Weaknesses:**

The main reason for me recommending that the paper is rejected is that I perceive the overall approach to be flawed. Given that the length scale selection scheme doesn't depend on the actual observed value (it only depends on the maximum distance between two training observations, the number of datapoints and the value of the regularisation parameter), it does not really qualify as an estimator but rather as a rule of thumb. Good rules of thumb can definitely be of great value to practitioners but I would then expect a much more extensive empirical validation on why this procedure is good, and what are the expected limitations.

The two main claims of the paper are that the method is computational inexpensive and that there is little variation in the length-scale estimates when randomising the experiment. I find these two claims to be very weak since an estimation method that returns, say, a length-scale of 1 would nail these two criteria. I would have appreciated if the focus were more centred on why the proposed selection scheme yields good estimates (or models), ideally with an ablation study before and after the approximation.

Given that the estimator doesn't depend on the actual observed value, the experimental results showing that the method outperforms marginal likelihood estimation appear suspicious to me, in the sense that I would expect this to be the exception and not the rule. The authors do mention that maximum likelihood may suffer from local minima, but could they be more explicit on how the optimisation was initialised?

In my opinion, these limitations do not make the paper publishable. I thus have not checked the math in details.

**Questions:**

Q1. The field of hyper-parameter estimation is much more developed in the Gaussian process regression setting. Why did the author choose the KRR setting instead?

Q2. Would it be possible to test the proposed approach on GP samples? With such settings there is a ground truth value for the length scale parameters that can be used to compare various methods. Such experiment could convince me of the performance of the approach if the proposed selection scheme isn't too far away from maximum likelihood. Ideally such an experiment would also highlight the limitations.

Q3. Maximum likelihood estimation is a principled way to estimate parameters, and typically outperforms cross validation unless there is some model specification (https://hal.science/hal-00905400/document). Since you observe this too on your experiments, does it mean that the Gaussian kernel is actually poorly suited for the data at hand?

Q4. My understanding is that the regularisation parameters and the kernel variance are not estimated in the experiments. How would that impact the results?



Remark 1.  Although it is more popular in geostatistics rather than machine learning, the (semi)-variogram method is significantly cheaper alternative to maximum likelihood or cross validation. Would suggest to include it to the benchmarks.

Remark 2. The exposition around $l_{max}$ could be improved. The definition kind of changes between a maximum distance to a median of the distance to the closest neighbour, it is thus slightly unclear what is recommended procedure in the end.

---

### Official Review · Reviewer_qguk · 2023-11-10

**Soundness:** 3 good
**Presentation:** 3 good
**Contribution:** 2 fair
**Rating:** 5
**Confidence:** 4

**Summary:**

The paper proposes a bandwidth selection procedure for the Gaussian kernel in kernel ridge regression using Jacobian regularization. To tackle this, a uniform approximation is computed to the norm of the Jacobian over the input domain.  It can be decomposed as the product of two components: one that controls the conditioning of the data kernel matrix, and one that controls how quickly the function decays to 0 away from data. Its behaviour is investigated and three different phases are considered and (local) minimum is analytically computed for each phase when it exists. The method is compared on KRR on three real-life and one synthetic dataset against cross-validation, maximum likelihood and Silverman's method. The Jacobian method is shown to be more efficient and more stable compared to the considered alternatives.

**Strengths:**

_Clarity_: The presentation is clear and easy to follow.

_Novelty_: The proposed approach is novel, and builds on previous ideas for Jacobian regularization.

_Significance_: As Gaussian KRR is still a fairly popular method, fast hyperparameter selection can be interesting.

_Analysis_: A detailed analysis is provided for the proposed approximation, and its behaviour is well-understood under various regimes.

_Visualizations_: Ample visualization is provided to guide the reader in understanding the objective components, and the breakdown of the results

**Weaknesses:**

_Heuristic_: The main idea to control the Jacobian is mostly heuristic-based with no rigorous theory supporting its use in practice.

_Experiments_: The experiments are mostly small scale and only consider toy datasets; scalability is not investigated. For example, it would be interesting to know whether the proposed approach performs well in scalable kernel methods, such as RFF or Nystrom with ridge regression. As datasets get large, fitting the full KRR becomes prohibitively expensive, and these are the settings where efficient bandwidth selection also becomes increasingly important. The baselines considered are somewhat lacking, for example, standard 3- or 5-fold CV is not considered, which according to my knowledge is the gold standard, or the median heuristic, which is known to provide good performance at low computational cost.

_Limited_: The application of the method is rather limited so far, since only the Gaussian kernel is considered. Although extensions to other kernels are mentioned to be possible, further calculations have to be carried out to derive the approximation.

**Questions:**

- When are the bounds eq. (8) and (9) expected to work well? What assumptions are we making about the distribution of training data?
- Is the approach robust  to using only a small subset of the dataset when the dataset is very large?
- On Figure 3, where is the true Jacobian evaluated? Is it an average over the training points?
- On Figure 6, what causes the instability of MML in the middle column?
- Similarly, in Table 1 what causes the underperformance of the MML approach?
- Can we compare the learned sigma value compare to the one by median heuristic?